# Extraction of Information on Trees outside Forests Based on Very High Spatial Resolution Remote Sensing Images

**Bin Sun** [1,2]**, Zhihai Gao** [1]**, Longcai Zhao** [3] **, Hongyan Wang** [3,*] **, Wentao Gao** [3] **and Yuanyuan Zhang** [1]

[1]    Institute of Forest Resource Information Techniques (IFRIT), Chinese Academy of Forestry (CAF),
       No. 1 Dongxiaofu, Haidian District, Beijing 100091, China; sunbin@ifrit.ac.cn (B.S.); zhgao@ifrit.ac.cn (Z.G.);
       zhangyuanyuan@ifrit.ac.cn (Y.Z.)
[2]    European Space Research Institute (ESRIN), European Space Agency (ESA), Via Galileo Galilei,
       00044 Frascati, Italy
[3]    Institute of Remote Sensing and Digital Earth (RADI), Chinese Academy of Sciences (CAS),
       Chaoyang District, Beijing 100094, China; zhaolc@radi.ac.cn (L.Z.); 18612534826@163.com (W.G.)
*    Correspondence: wanghy@radi.ac.cn; Tel.: +86-10-64837645

**Abstract:** The sparse *Ulmus pumila* L. woodland in the Otingdag Sandy Land of China is indispensable in maintaining the ecosystem stability of the desertified grasslands. Many studies of this region have focused on community structure and analysis of species composition, but without consideration of spatial distribution. Based on a combination of spectral and multiscale spatial variation features, we present a method for automated extraction of information on the *U. pumila* trees of the Otingdag Sandy Land using very high spatial resolution remote sensing imagery. In this method, feature images were constructed using fused 1-m spatial resolution GF-2 images through analysis of the characteristics of the natural geographical environment and the spatial distribution of the *U. pumila* trees. Then, a multiscale Laplace transform was performed on the feature images to generate multiscale Laplacian feature spaces. Next, local maxima and minima were obtained by iteration over the multiscale feature spaces. Finally, repeated values were removed and vector data (point data) were generated for automatic extraction of the spatial distribution and crown contours of the *U. pumila* trees. Results showed that the proposed method could overcome the lack of universality common to image classification methods. Validation indicated the accuracy of information extracted from *U. pumila* test data reached 82.7%. Further analysis determined the parameter values of the algorithm applicable to the study area. Extraction accuracy was improved considerably with a gradual increase of the Sigma parameter; however, the probability of missing data also increased markedly after the parameter reached a certain level. Therefore, we recommend the Sigma value of the algorithm be set to 90 (±5). The proposed method could provide a reference for information extraction, spatial distribution mapping, and forest protection in relation to the *U. pumila* woodland of the Otingdag Sandy Land, which could also support improved ecological protection across much of northern China.

**Keywords:** *Ulmus pumila* sparse forest; Otingdag Sandy Land; tree detection; crown extraction; GF-2 data; automated extraction algorithm

---

## 1. Introduction

Globally, China is one of the countries most affected by desertification. In 2015, China's State Forestry Administration published *A Bulletin of Status Quo of Desertification and Sandification in China*, which indicated that, as of 2014, the areas of desertified and sandy land in China were 2,611,600 and

1,721,200 km$^2$, respectively [1]. Although these figures have decreased in the past three decades, the prevention of further desertification remains a challenge. Trees outside forests (TOFs) are an important natural resource that contributes substantially to national biomass and carbon stocks and to the livelihood of people in many regions [2,3]. However, TOFs are generally absent from national forest inventories [4], and this lack of data precludes a proper assessment of their contribution to landscape connectivity with regard to associated species [5]. As one type of distribution of TOFs, the *Ulmus pumila* woodland found in sandy land is not a forest in the true sense of the word because *U. pumila* trees grow sparsely in such environments with an understory of shrubs or herbs, presenting the characteristics of grasslands rather than typical forest [6]. The characteristics of *U. pumila* L. include a strong root system, small transpiration intensity, wide crown, photophilous, and cold and drought resistance [7,8]. With its unique vegetation features, the sparse *U. pumila* trees that are widespread within the Otingdag Sandy Land and its surroundings play important roles as a windbreak, in climate regulation, and in the maintenance of ecosystem stability of the desertified grasslands [9,10]. Therefore, studies on techniques for identification of individual *U. pumila* in the Otingdag Sandy Land are of scientific and ecological importance, it is the base of a series of studies, which have mainly focused on *U. pumila* community, structure analysis, population distribution pattern, and water conservation function assessment in sandyland. In addition, such techniques could help both in delineating the spatial distribution of *U. pumila* within the region and in developing countermeasures designed to maintain and improve the regional ecosystem stability and the implementation of sand prevention and control project in the Otingdag Sandy Land.

In comparison with the extensive amount of research that has been conducted on individual tree identification and crown extraction of savanna woodland in Africa, Australia, India, and the United States [11–14], there have been few studies on the sparse *U. pumila* woodland in the Otingdag Sandy Land of China. Most previous related research has focused on community structure [8,15] and species composition [16,17], and only a small number of studies have reported on the spatial distribution of individual trees and the extraction of crown shapes. Earlier studies that have considered techniques for locating individual trees or extracting crown information have tended to focus on areas where forest vegetation is abundant [18,19].

Conventional extraction techniques include field surveys and literature reviews; however, these time-consuming and laborious methods are restricted by observational methods and data availability. Moreover, it is difficult to map large-scale spatial distributions of individual trees using such techniques. Following recent advances in remote sensing and computing technologies, new technical means and data sources have become available for forestry resource surveying and automated information extraction. For example, remote sensing data can be acquired using satellite-based optical sensors [20–22], LIDAR [23–26], unmanned aerial vehicles [27–29], or a combination of multiple data sources [30].

Optical remote sensing data suitable for large-scale mapping can be obtained easily across a wide observation area. However, the low- and moderate-resolution data captured by the Landsat, Terra, and Aqua satellites are unable to meet the needs of information extraction of individual trees. Therefore, very high resolution remote sensing data are used more widely for such purposes. These data include the 0.61-m panchromatic data collected by QuickBird, 1-m panchromatic data collected by IKONOS, 1.5-m panchromatic data collected by Spot-6, 0.5-m panchromatic data collected by Worldview-2, 5-m panchromatic data collected by CBERS-04, 2.1-m data acquired using the orthographic camera onboard the ZY-3 satellite, and 0.5-m panchromatic data collected by SuperView-1. However, the acquisition of very high resolution remote sensing data can be expensive because swath limitations can mean huge costs in the monitoring of large regions. On August 19, 2014, China successfully launched the GF-2 satellite as a platform for two 1-m panchromatic and 4-m multispectral cameras. This is China's first self-developed optical remote sensing satellite (spatial resolution: ≤1 m) designed for civilian purposes. The spatial resolution at the nadir point can reach 0.8 m [31], which means forestry is one potential area of application for GF-2 data. To date, GF-2 data have been used widely in major forestry tasks, e.g., forest resource surveying and monitoring, desertification, and rocky desertification monitoring,

wetland resource surveying and monitoring, forest ecological engineering monitoring, and forest pest monitoring. Therefore, the study of methods using GF-2 data for individual identification of *U. pumila* trees in sandy lands is of considerable practical importance for the sake of broadening the application of domestic remote sensing data.

The crown of individual trees can be identified clearly using very high resolution remote sensing imagery, and from visible features such as geometric patterns and shapes, the tree crown diameter can be extracted accurately. Currently, most methods used for locating individual trees and extracting tree crown information can be classified under the following two categories: image-classification-based extraction and target-identification-based extraction. Of the various image-classification-based extraction methods [32–34], the object-oriented classification method is widely used [35,36]. By considering spatially connected pixel sets (objects) with similar spectra as processing units, the method performs computerized classification of segmented images to realize crown information extraction and spatial location identification. The primary advantage of this method is that it can make full use of data to achieve high extraction accuracy through parameter tuning. However, one disadvantage is that there are often too many variable parameters to be set. Because land features of single-scene images vary, different scales and categorization variables must be selected each time the processes of segmentation and classification are performed. Consequently, the method is unsuitable for mapping the large-scale spatial distribution of individual trees. In target-identification-based extraction methods [37–39], computer algorithms use a pixel-based method to extract features from the pixels surrounding a target identified in an image; thus, realizing the extraction of information of individual trees. Common extraction methods include valley following [11,40,41], local maxima detection [42–44], template matching [45–47], seeded region growing [48–50], and watershed segmentation [51–53]. Generally, such methods are highly adaptable, and they can meet the requirements for regional mapping. However, to achieve high-accuracy automated extraction, different algorithms must be designed depending on the environment in which the targets are located. Both the image features of the target and the spectral features of the surrounding objects should be considered. Therefore, to meet the needs of regional mapping in relation to the sparse *U. pumila* woodland in the Otingdag Sandy Land, a method that is effective in using Chinese remote sensing data for the extraction of information on individual trees is required.

An efficient scientific technique for the extraction of information regarding *U. pumila* will be conducive to delineating the spatial distribution of the sparse forest in the Otingdag Sandy Land, which is prerequisite for conducting further related scientific research. Therefore, this study considered the use of GF-2 imagery to investigate the sparse *U. pumila* trees in the eastern part of the Otingdag Sandy Land. A multiscale automated identification method was developed, based on fused image feature transform, to extract the locations of individual trees in different spatial distribution patterns. The findings of this study could provide support and reference for scientific decisions regarding the restoration, conservation, and management of land affected by desertification and sandification in the Otingdag Sandy Land.

## 2. Materials and Methods

### 2.1. Study Area

The study area is located in the eastern part of the Otingdag Sandy Land in the north of Zhenglanqi, Xilin Gol League, Inner Mongolia Autonomous Region, where the natural sparse *U. pumila* woodland is concentrated (Figure 1). The Otingdag Sandy Land is one of four major sandy lands in China. Of all the source areas of sandstorms that affect the Beijing–Tianjin area, it is the closest to Beijing. The area is severely desertified and sensitive to global climate change and thus it has great ecological importance. Following the implementation of the Beijing–Tianjin Sandstorm Source Area Control Project in 2000, the overall ecological environment of the area has improved, although local alternation of degradation and restoration remains [54,55]. The Otingdag Sandy Land has sparse coverage of typical grassland and sandy land vegetation. The trees are predominantly *U. pumila* that grow in

isolation or in small groups in central and eastern areas. Their spatial distribution is reasonably small and uneven, forming a typical landscape of sparse *U. pumila* trees. Therefore, determining the total number and delineating the spatial distribution of *U. pumila* trees in the Otingdag Sandy Land is useful for developing appropriate protection policies for ecological stability and sustainable development of the region, such as the determination of the core ecological engineering implementation area of the sand control project and the formulation of national ecological subsidies.

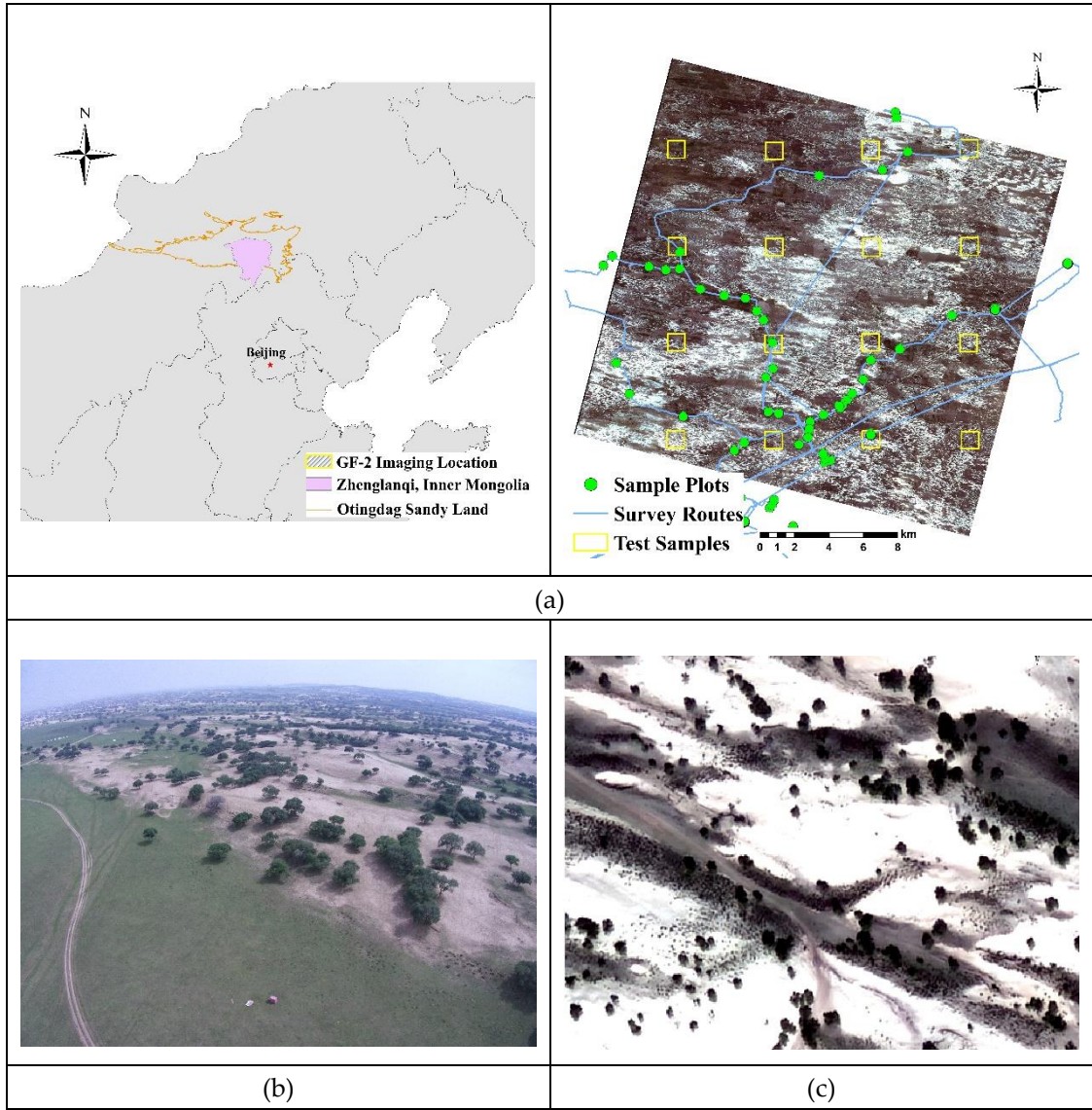

**Figure 1.** Typical *Ulmus pumila* landscape and field survey in the study area. (**a**) Study area, GF-2 imaging location and field survey; (**b**) *Ulmus pumila* landscape in the hinterland of the Otingdag Sandy Land captured by an unmanned aerial vehicle; and (**c**) *Ulmus pumila* landscape in GF-2 imagery (fused image, 1-m resolution).

## 2.2. Data Collection and Processing

### 2.2.1. Remote Sensing Data

The remote sensing data used in this study were acquired by China's GF-2 satellite, which has two cameras, one that provides 1-m spatial resolution panchromatic (0.45–0.9 um) and one that provides 4-m resolution multispectral. The multispectral data cover four bands: blue (0.45–0.52 μm), green (0.52–0.59 μm), red (0.63–0.69 μm), and near IR (0.77–0.89 μm). Field surveys have found that the

period from mid-July to the end of August is the growing season of ground vegetation in the study area, which is when the grassland biomass reaches its peak. To reduce noise interference from other land features during extraction of individual tree location and crown information, the extraction of *U. pumila* information should preferably be based on GF-2 imagery collected after germination (end of May to beginning of July) or before the fall of leaves (September–October). This study used images captured on September 16, 2016 (Figure 1b).

The FLAASH model was adopted for radiation calibration and atmospheric correction using a radiation calibration coefficient provided by the China Centre for Resources Satellite Data and Application (http://www.cresda.com/). Geometrically corrected ZY-3 data were used as reference for the geometric correction. No fewer than 50 evenly distributed control points were selected, and the error was acquired to within 1 pixel. Using the nearest-neighbor diffusion pansharpening method, 4-m multispectral data from GF-2 were fused with 1-m panchromatic data to generate a new 1-m multispectral image for the extraction of the location and crown shape of individual *U. pumila* trees. Figure 2 shows partial enlargements of the original multispectral image, original panchromatic image, and fused image.

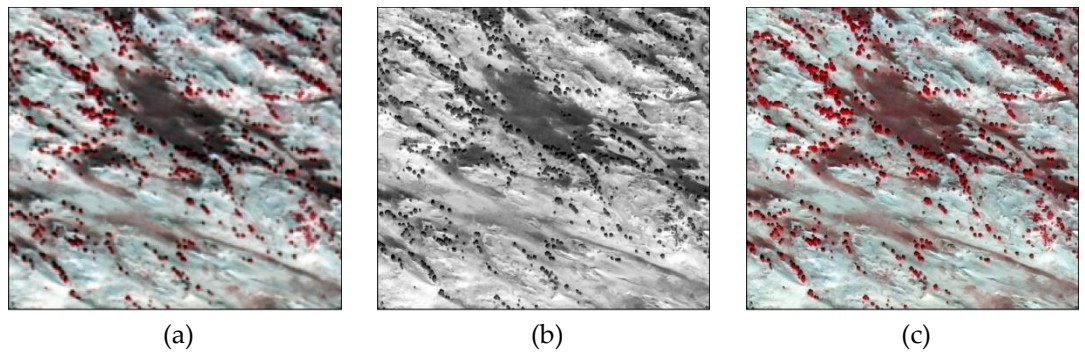

|     |     |     |
| :-: | :-: | :-: |
| (a) | (b) | (c) |

**Figure 2.** Comparison of GF-2 panchromatic, multispectral, and fused images (partial). (**a**) Original multispectral image (composed with red, greed, and blue bands), (**b**) original panchromatic image, and (**c**) nearest-neighbor diffusion fused image (composed with red, greed, and blue bands).

### 2.2.2. Field Survey Data

The research team has conducted field surveys in Otingdag and its surrounding areas annually during July–August since 2011, and it has accumulated a large volume of ground survey data. During 2014–2016, three plot surveys were undertaken that involved spectral measurements, individual tree measurements, and high-accuracy GPS positioning of *U. pumila* specimens. Overall, location information and growth status indicators of 601 individual *U. pumila* trees were obtained.

To verify the extraction results obtained using the algorithm on the GF-2 image, we set 16 uniform test quadrats, arranged in a 4 × 4 grid, each measuring 1 × 1 km (see Figure 1a). Based on the fused GF-2 image, Google Earth, and very high spatial resolution UAV data, the location and crown of 2913 individual *U. pumila* trees were obtained by visual interpretation and artificial delineation. Data from the field surveys and the test quadrats were used to verify the extraction result. The accuracy of the extraction result was evaluated using Equation (1):

$$\delta = a / b \tag{1}$$

where *b* is the total number of extracted trees, *a* is the number of correctly extracted *U. pumila* trees through verification, and δ is the detection accuracy.

*2.3. Method*

The spatial resolution of the fused GF-2 images was 1 m; thus, the contour of each tree crown could be observed clearly. However, such high resolution also introduces more within-class variance to the extraction of the study target. Therefore, to realize automated identification of the location and crown of individual trees, it is necessary to enhance the information of each specimen and to suppress other information. For this purpose, feature images were constructed using the fused GF-2 images. Then, different window sizes of Gaussian filters was performed on the feature images to generate multiscale feature spaces. Next, Laplace transform was applied on the multiscale feature space to generate multiscale Laplacian feature spaces, and local maxima and minima were obtained by iteration over the multiscale Laplacian feature spaces. Finally, repeated values were removed and vector data (point data) were generated to obtain the crown contours and to produce a map of the spatial distribution of *U. pumila*. The flowchart of the main technical process is shown in Figure 3.

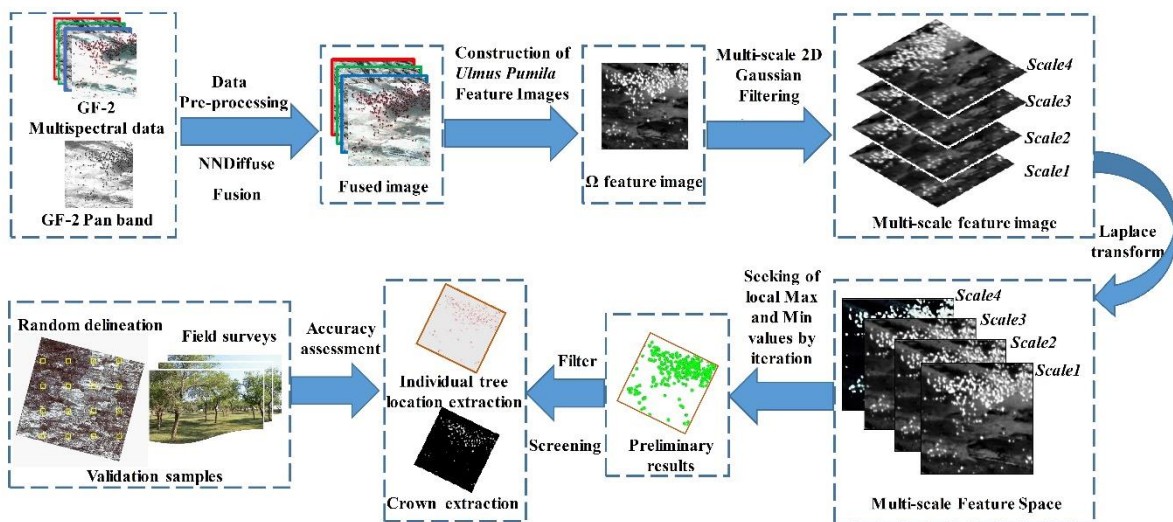

**Figure 3.** Flowchart of the automated process for extraction of information on sparse *Ulmus pumila* woodland in the Otingdag Sandy Land.

2.3.1. Construction of *U. pumila* Feature Images

In the GF-2 image of the study area, the vegetation is distributed sparsely and uniformly, and the soil texture is obvious; therefore, the image shows distinct changes in the normalized difference vegetation index (NDVI) of soil, grassland, shrubs, and *U. pumila*. Based on the analysis of remote sensing features of ground objects within the study area, we first considered constructing NDVI-dominated $\Omega$ feature images [56] for *U. pumila* patch detection. The $\Omega$ calculation is shown in Equation (2) [57]:

$$\Omega = \frac{\pi}{4}\arctan(NDVI) \tag{2}$$

$$NDVI = (NIR - Red)/(NIR + Red) \tag{3}$$

where NDVI is the normalized difference vegetation index, NIR is the near IR band reflectance, and Red is the red band reflectance. The construction of the preliminary *U. pumila* feature image with an NDVI nonlinear extension can alleviate the problem of NDVI saturation of lush trees. After processing, the *U. pumila* targets become bright patches in the feature image, while other object features are suppressed and darkened.

2.3.2. Construction of Multiscale Feature Space

The very high spatial resolution image makes visual interpretation and information extraction more intuitive; however, it also introduces more noise that affects target extraction. To eliminate noise

interference, the generated NDVI-dominated $\Omega$ feature images in the first step must be smoothed to facilitate the extraction of *U. pumila* location information.

The sparse *U. pumila* woodland in the Otingdag Sandy Land has obvious distribution features that include single trees, contiguous distribution of a few trees, and contiguous distribution of many trees, which indicate that various types of extraction accuracy requirement cannot be satisfied at the same scale. Therefore, the construction of multiscale feature spaces was considered to achieve accurate information extraction of *U. pumila* with different distribution forms.

The Gaussian filter is a linear smoothing filter that is applied widely for noise reduction during image processing. In this process, a pixel-level weighted average strategy is used. The value of each pixel is calculated as a weighted average of the pixel and its neighboring pixels, which can effectively suppress normally distributed noise. Therefore, in combination with the feature image constructed following the method in Section 2.3.1, a Gaussian filter was applied to the feature image to generate a multiscale feature space using different filter kernel sizes and sigma combinations. The equation for the 2D Gaussian filter is [58]:

$$G(x, y) \ = \ \frac{1}{2\pi\sigma^2}e^{-\frac{x^2+y^2}{2\sigma^2}} \tag{4}$$

where $x$ and $y$ represent the distance between the central pixel and its neighbors, and $\sigma$ represents the standard deviation. In the multiscale feature space, the Gaussian smoothed bright patches in the feature images at different scales represent different sizes and different distribution types of *U. pumila*. In this study, the window size of Gaussian filters was set as 3, 5, 7, and 9, respectively; thus, 4 kinds of different scale feature spaces were constructed.

### 2.3.3. Approximate location of the *U. pumila* targets detection

Next, to extract details of the crown of *U. pumila* trees, the Laplace transform should be applied to the obtained multiscale feature spaces. The Laplace operator is a second-order differential operator in n-dimensional Euclidean space, defined as the divergence of the gradient of a spatial scalar function. It is used widely in image processing techniques such as image sharpening and edge detection. The Laplacian is invariant under all Euclidean rotations. In a 2D image, the Laplace transform of a function is an isotropic measure of the second spatial derivative of the image. The Laplace operator in the Cartesian coordinate system is defined as [59]:

$$\text{Laplace}(f) \ = \ \frac{\partial^2 f}{\partial x^2} + \frac{\partial^2 f}{\partial y^2} \tag{5}$$

Using the Laplace operator on the multiscale feature spaces, the center of each bright patch at different scales was detected, i.e., determining the approximate location of the *U. pumila* targets. In addition, unexpected bright patches with oversized or undersized area were removed by defining thresholds for the maximum and minimum area.

### 2.3.4. Location Identification of *U. pumila* Trees and Crown Detection

Finally, by searching spread over the multiscale feature spaces operated by Laplacian, every local maximum value point (i.e., possible *U. pumila* target point) at each scale could be determined and saved as a candidate *U. pumila* target. The radius of each candidate *U. pumila* target was set at $r = 1.5 \times \sigma$. The equation for the standard deviation ($\sigma$) is:

$$\sigma = 1 - \frac{\text{Sigma}}{100} \tag{6}$$

where the value of Sigma is 0–100. Because of the possibility of serious overlapping of candidate targets found at different scales, candidate *U. pumila* targets with large overlapping areas were deleted, and the remaining target was retained as the detected *U. pumila* crown contour. Next, the center of the

smallest circumscribed rectangle of the patch was calculated as the center of the patch, i.e., the location of an individual tree of the sparse *U. pumila* woodland.

In summary, by applying the four-step image processing and extraction algorithm mentioned above, the position of *U. pumila* trees and their crown can be extracted effectively.

## 3. Results and Analysis

### 3.1. Parameter Analysis and Setting

In this method, two parameters must be determined: the patch size for extracting different distribution types and Sigma, as mentioned in Section 2.3.4.

#### 3.1.1. Patch Size (Patch_min and Patch_max)

The values of Patch_min and Patch_max are closely related to the diameter of the clumps of crowns of *U. pumila*. As described in Section 2.3.4, the smaller the value of Patch_min, the greater the number of small patches included in the results, and vice versa. The larger the value of Patch_max, the greater the number of large patches included in the results, and vice versa. To determine suitable parameters applicable to the study area, we summarized the east–west and south–north crown diameters of more than 600 trees measured during the field surveys, as shown in Figure 4.

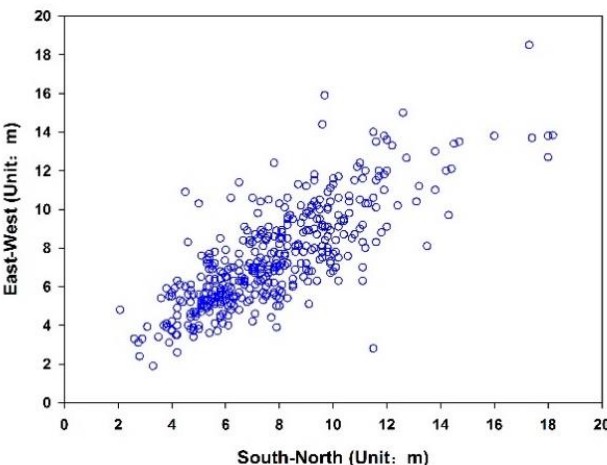

**Figure 4.** Scatter diagram for *Ulmus pumila* crown diameters measured during field surveys of the study area.

It was found during the field surveys that the size of the crown of individual *U. pumila* trees was 2–20 m. Thus, in the study area, the projected area of an individual tree in the image was approximately 4–400 m$^2$. Considering the representativeness of the samples and that crowns that overlap are indistinguishable because of the close tree arrangement, the Patch_max value should be increased. Therefore, through analysis of the spatial distribution features of *U. pumila* crowns within the study area, we determined the thresholds for filtering the patches of Laplace transformed multiscale space, i.e., Patch_min was set to 5 m$^2$ and Patch_max was set to 1000 m$^2$. This is to find a balance point to avoid missing detection and misjudgment as much as possible.

#### 3.1.2. Sigma Value

Sigma is one of the most important parameters affecting the extraction accuracy of *U. pumila*. Section 2.3 suggests that Sigma will not only affect the final extraction accuracy but also the missing detection and false detection rates. Therefore, we set Sigma to values of 60, 65, 70, 75, 80, 85, 90, 95, 97, and 99, and substituted them in turn into the algorithm to output detection results. The results were then compared with the verification data mentioned in Section 2.2.2. Next, we summarized the

number of correctly extracted *U. pumila* trees, the accuracy rate, and the effective extraction rate of the test samples, analyzed the relationships between Sigma and the accuracy and effective extraction rate, and determined the value of Sigma applicable for the images of the study area. This also provided a reference for the setting of other regional parameters applicable to the study area. The results are shown in Figure 5.

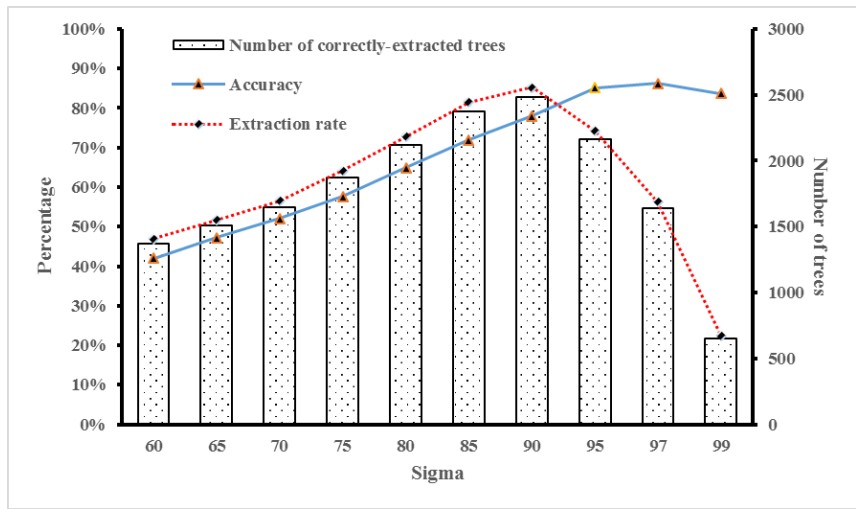

**Figure 5.** Variation of extraction accuracy with change of Sigma value.

Figure 5 shows that with the increase of Sigma, the extraction accuracy for *U. pumila* trees gradually increases and becomes reasonably stable. For example, when Sigma is 60, the extraction accuracy is only 42.0%. However, when Sigma is 95 and 97, the extraction accuracy is 85.1% and 86.3%, respectively. Initially, as Sigma increases, the number of correctly extracted trees in the samples gradually increases but then it starts to fall back after reaching a specific value (Sigma = 90). When Sigma is 60, only 1370 of 2931 trees in the samples are extracted correctly, i.e., an extraction rate of 47.0%. When Sigma is 90, the extraction rate increases to 85.3% and 2484 trees are extracted correctly. When Sigma is >90, the extraction rate goes down. When Sigma is 97, the extraction rate is just 22.5% and only 654 trees are extracted correctly. This indicates that with a gradual decrease of the standard deviation, the model imposes a stricter and more cautious assessment of whether the target it is an *U. pumila* tree. Although the correct extraction rate increases significantly, the number of missed *U. pumila* trees also increases. Conversely, when Sigma is small (i.e., when the standard deviation is large), the corresponding missing detection rate is much lower, but the error rate of the extraction results increases accordingly. Based on the above, a large value of Sigma is not always optimal. For larger values of Sigma, the extraction accuracy and the missing extraction rate will both increase. Therefore, it is necessary to establish a balance. From the above analysis, it is recommended that Sigma be set to 90 (±5).

To present the relationships between Sigma and the extraction effect more intuitively, we selected a partial area from the image and superimposed the extraction results of different Sigma values onto the image, as shown in Figure 6. It can be seen from the figure that the number of extracted trees gradually increases when Sigma increases from 60 to 90. When Sigma is 60, the level of incorrect assessment is obvious. Then, it gradually improves with the increase in the value of Sigma. When Sigma is 90, most *U. pumila* locations are extracted accurately. However, with further increase in the value of Sigma, the number of trees extracted correctly decreases significantly. In particular, when Sigma = 99, only five *U. pumila* trees are extracted correctly. For Sigma values of 60 and 97, the total number of extracted trees is similar but the extraction accuracy at Sigma = 97 is significantly higher than that at Sigma = 60. This further indicates that the extraction accuracy rate gradually increases with the gradual increase of Sigma, but after the Sigma value exceeds the balance point, the missing detection rate will also increase significantly.

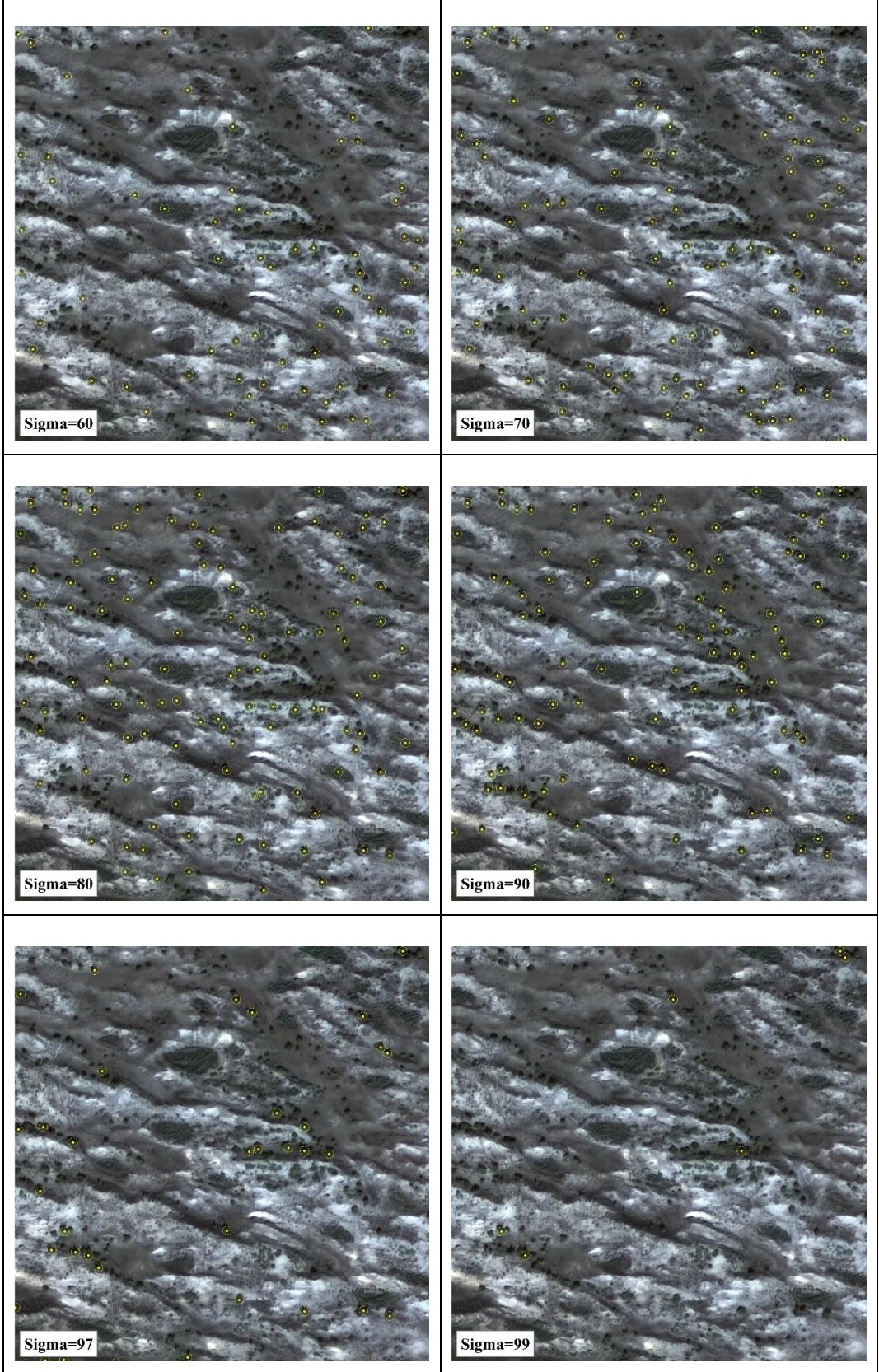

**Figure 6.** Extraction at different Sigma values (partial).

### 3.2. Image Extraction Result

Based on analysis of the results in Section 3.1, we determined the key parameters in the algorithm, and then we extracted the spatial distribution and crown scope of the *U. pumila* trees from the image. The results are shown in Figure 7.

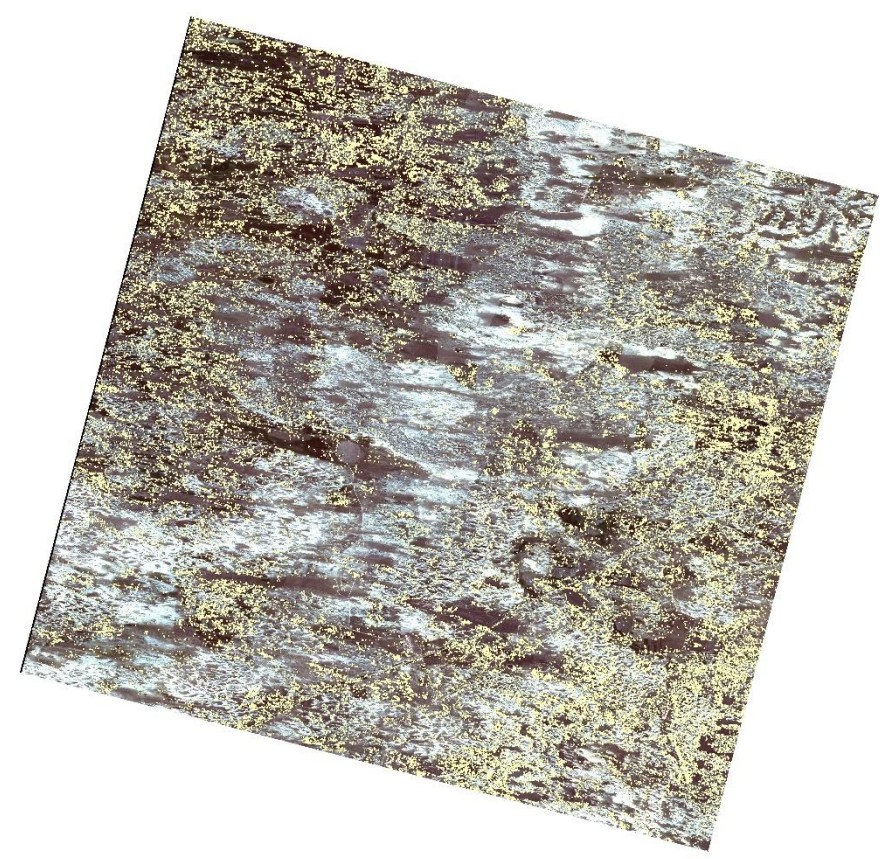

**Figure 7.** Spatial distribution of sparse *Ulmus pumila* trees based on fused GF-2 data.

As could be seen in Figure 7, the *U. pumila* trees in the fused GF-2 image are distributed mainly in areas of fixed sandy land, semi-fixed sandy land, and flat bottomland, with a scattering of trees in the moving sandy land with obvious drift sand. Verified by test samples, the extraction accuracy is 82.7% and the overall extraction rate of *U. pumila* samples is 83.4%. Figure 8 shows the effect of local extraction. Most *U. pumila* trees are identified effectively and their crown scope is largely consistent with actual measurements. Especially for single trees and contiguous distribution of a few trees in the sandy land, the extraction effects are satisfactory (Figure 8a,b) because of the high contrast between the *U. pumila* specimens and the background. The degree of heterogeneity is significant because after a series of image changes, the *U. pumila* information has been enhanced markedly, while the background information has been suppressed. However, for contiguous distribution of many trees in the sandy land, the surface vegetation is also lush. Therefore, the missing extraction rate is higher than that of the single-tree distribution. As shown in Figure 8c,d, the specimens of *U. pumila* within the red boxes have small crown diameters. In the bottomland, which has strong water-retaining capacity, surface vegetation grows vigorously such that the *U. pumila* crowns are mixed with the surface vegetation, which leads to failure of effective extraction of the crown contours. For contiguous areas of many *U. pumila* trees with large crown diameters, as shown in the blue boxes of Figure 8c, the crowns of some individual *U. pumila* specimens are masked to a certain extent by the crowns of adjacent trees. Therefore, the algorithm fails to extract the location of individual trees accurately; instead, the crowns are extracted in contiguous form.

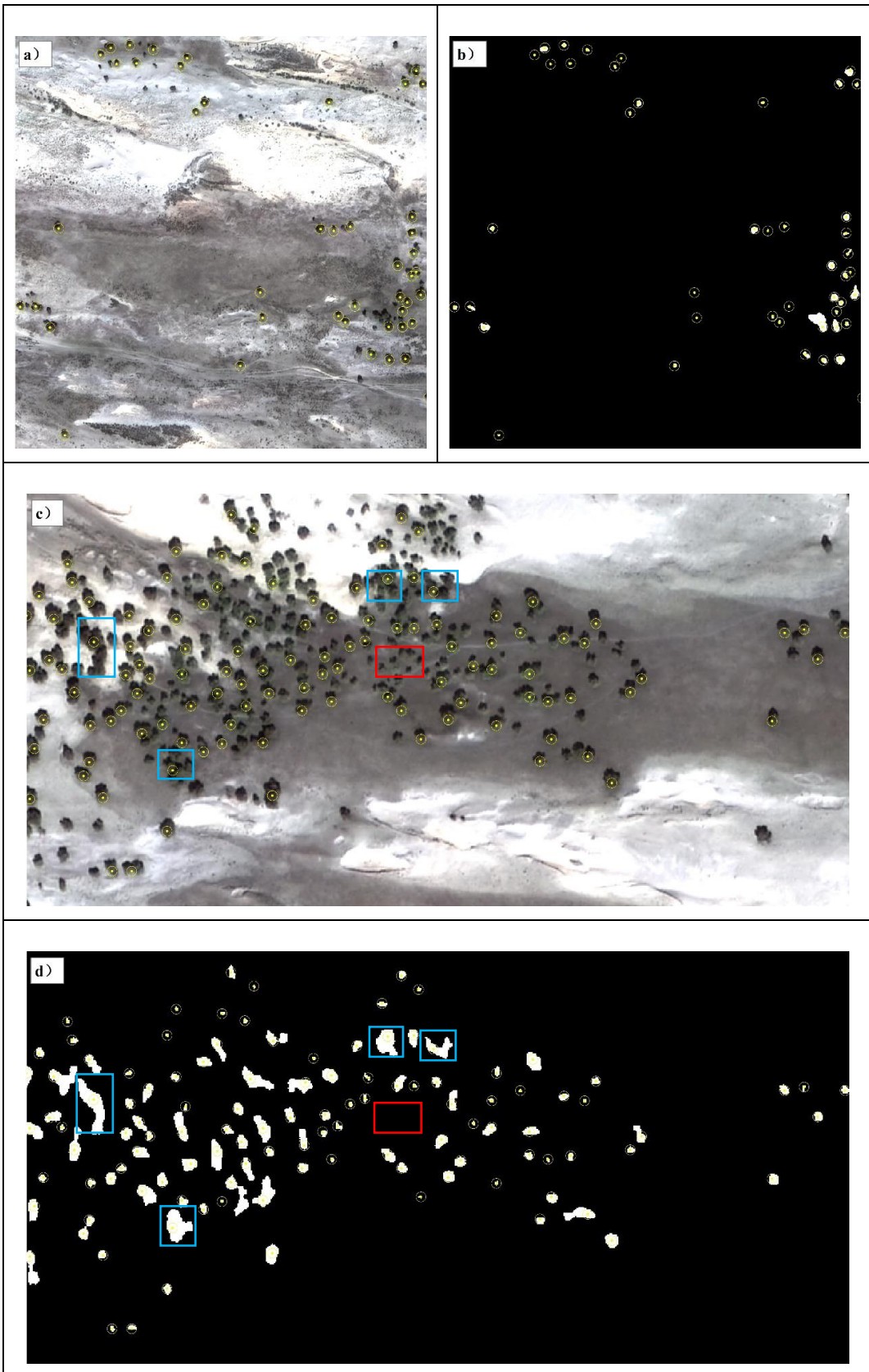

**Figure 8.** Extraction effect of sparse *Ulmus pumila* trees in sandy land (partial). (**a**,**b**) are the location and tree crown extraction results of *Ulmus pumila* in areas with single trees and contiguous distribution of a few trees; (**c**,**d**) are the location and tree crown extraction results of *Ulmus pumila* in areas with contiguous distribution of many trees.

### 3.3. Comparison with the Image-Classification-Based Extraction Results

Currently, because of the cost of observations and other limiting factors, few studies have investigated the extraction of information on sparse *U. pumila* trees in the Otingdag Sandy Land using very high resolution remote sensing data. To verify the extraction results of the method proposed in this study, we compared the extraction accuracy we achieved with the results from the widely used supervised classification methods, e.g., the Maximum Likelihood, Mahalanobis Distance, Support Vector Machine, and Neural Net methods. As described in the introduction section, the four methods on behalf of the "image-classification-based extraction method" and the method we proposed in this study represents the "target-identification-based extraction method". All of the parameters in the four methods were all determined and adjusted to achieve the optimal state of the classification results, and then the classification results were achieved. In addition, data for training and validation were the same; that is, 16 test quadrats were divided into two parts: 11 for training and 5 for validation. The classification accuracy is shown in Table 1.

**Table 1.** Identification accuracy of *Ulmus pumila* trees using remote sensing image classification methods.

| Methods | Producer's Accuracy (%) | User Accuracy (%) |
|---|---|---|
| Maximum Likelihood | 74.91 | 33.52 |
| Mahalanobis Distance | 63.61 | 37.39 |
| Neural Net | 58.7 | 66.79 |
| Support Vector Machine | 73.2 | 72.18 |

From the statistical results, we found that although the classification accuracy of *U. pumila* trees achieved by the maximum likelihood method was highest (reaching 74.9%), the user accuracy was only 33.52%. The Support Vector Machine showed relatively satisfactory performance with 73.2% classification accuracy and 72.18% user accuracy. Overall, comparing the results we achieved based on the proposed target-identification-based extraction method, the commission and omission errors were more serious among the four supervised classification methods. Through visual evaluation of the classification results, it was found that some *U. pumila* shadows, tall shrubs, and herbaceous wetlands were misclassified as *U. pumila* trees. Using the method proposed in this study, the extraction accuracy using the same test data reached 82.7% with a relatively high extraction rate. Therefore, the comparison verification results suggest that compared with traditional image classification methods, the proposed algorithm is effective in reducing misjudgments and missed judgments, while improving the identification accuracy of *U. pumila* trees.

## 4. Discussion

The method we proposed in this study could be discussed and improved further in terms of the following aspects:

(1) The method has good universality. In our study, based on the spectral features analysis at pixel scale, we proposed an effective method for extraction of information on TOF by using very high spatial resolution remote sensing images. From the perspective of wave bands, existing remote sensing data are shown to meet the extraction requirements of the study area; therefore, the method could be extended for use with other types of very high resolution data. From the perspective of data sources, two factors affect the extraction effect. One is the resolution of the image. It is obvious that if the spatial resolution of remote sensing images is lower than the size of the individual tree crown, the basis for *U. pumila* feature extraction was lacking. Therefore, the main data source for the proposed method should comprise very high resolution remote sensing data. At present, most very high resolution data are acquired by commercial satellites and thus are expensive to obtain. The GF-2 satellite is a platform that acquires high resolution imagery that is made available to the public; therefore, methods developed to use these easy access data could have a promising future. The selection of the time phase of the remote sensing image is the second important factor that affects the extraction effect. The core of

the method is to enhance the information of *U. pumila* crowns and to suppress other ground object features. Vegetation in desertified grasslands comprises mostly annual vegetation. Especially in years when the environmental conditions of water and heat are conducive, vegetation grows vigorously during the growing season. In such a circumstance, the surface vegetation tends to mix with the *U. pumila* crowns. This makes identification of crown boundaries difficult and it affects the extraction effect. For the period from late autumn to early spring, the features of leaf-off trees in remote sensing imagery are similar to exposed soil, which is not conducive to extraction. Therefore, for the proposed method, it is preferable to select GF-2 images acquired during the period after germination (end of May to beginning of July) or before the fall of leaves (September–October) to avoid interference of other ground object features. In the process of extraction of information on *U. pumila* crowns, we performed a series of feature transforms on the images to enhance crown information and suppress other ground object features. Despite this processing, the method remains ineffective for identification of large contiguous areas of *U. pumila* trees, especially in low-lying land between slopes where vegetation is lush. Therefore, large numbers of *U. pumila* trees are likely screened out because of interference from background vegetation. This further indicates the importance of the selection of time phase for correct extraction of *U. pumila* features.

(2) Promising application potential. The continuing enhancement of remote sensing data, especially very high spatial resolution remote sensing data, provides greater potential for the application of the proposed method. For example, the panchromatic and multispectral spatial resolutions of the three GF 1-02, 03, and 04 satellites in China's first natural resource monitoring constellation are 2 and 8 m, respectively. Furthermore, the GF-6 satellite, launched before the above three, is also equipped with a 2/8-m-resolution panchromatic/hyperspectral camera. The wave band settings of these satellites are similar to GF-2. Therefore, the proposed method could provide a reference for the application of other remote sensing images of equal or similar resolution. In particular, GF-6 is the first satellite to incorporate the "red edge" wave band that can effectively reflect vegetation features. It would be interesting to investigate the use of other spectral segments, especially to examine how best to fully reflect the vegetation feature spectrum for optimization of the algorithm model and for improvement of its computational efficiency. However, optical images reflect 2D information and the number of wave bands is limited. Therefore, future use of InSAR and LIDAR data, which can reflect height information, could help distinguish tree crowns from the top layer of surface vegetation. For example, in our study, it is difficult to separate trees in clumps. However, if we could identify individual trees first by optical data, and then further separate them in the key clumps distributed areas by adding the three-dimensional information of LIDAR. Maybe this would significantly improve the identification accuracy of *U. pumila* trees and increase crown extraction accuracy. However, a considerable challenge to the operational efficiency of the method should be expected if multisource data were incorporated.

## 5. Conclusions

The main conclusions derived from this study are as follows:

(1) Based on analysis of the spatial distribution features of the sparse *U. pumila* trees in the Otingdag Sandy Land, this study proposed a method for the extraction of crown and spatial location information of *U. pumila* trees based on fused GF-2 imagery and a multiscale spatial transform. The pixel-based method overcomes the shortcoming of common image classification methods. It was verified that the *U. pumila* extraction accuracy of the method using test data was up to 82.7%.

(2) Analysis indicated that the recommended values for filtering the smallest and largest patches within the study area are 5 and 1000, respectively. As the value of Sigma increased, the extraction accuracy improved significantly but the missing extraction rate also increased. Consequently, the recommended value of Sigma was set at 90 (±5).

(3) The method can realize automated extraction of crown and spatial location information of *U. pumila* trees without reference samples. Although this study focused on the use of GF-2 imagery, the proposed method could also be used with most other high-resolution remote sensing images, which

demonstrates its wide application potential. The findings of this study could provide a reference for further investigation of information extraction technology for application to the sparse forests found in sandy lands.

**Author Contributions:** Conceptualization, Z.G., B.S., and H.W.; Methodology, L.Z. and H.W.; Software, L.Z. and B.S.; Validation, B.S. and W.G.; Formal analysis, L.Z., Y.Z. and W.G.; Investigation, B.S., W.G., Y.Z. and H.W.; Resources, B.S. and Z.G.; Data curation, Y.Z. and W.G.; Writing—Original draft preparation, B.S. and L.Z.; Writing—Review and editing, Z.G., B.S., and H.W.; Visualization, B.S.; Supervision, Z.G. and H.W.

**Funding:** This research was funded by the Fundamental Research Funds for the Central Non-profit Research Institution of CAF, grant number CAFYBB2019ZB004, National Science and Technology Major Project of China, grant number 21-Y20A06-9001-17/18, National Natural Science Foundation of China, grant number 41501467, and ESA-MOST China Dragon 4 Cooperation, grant number 32396.

**Acknowledgments:** We would like to acknowledge Xiaolong Hu, Haiguang Huang and Limin Yuan of Inner Mongolia Academy of Forestry, China for their many years participating in field investigations. We also thank James Buxton from Liwen Bianji, Edanz Group China (www.liwenbianji.cn./ac), for editing the English text of this manuscript.

**Conflicts of Interest:** The authors declare no conflict of interest. The funders had no role in the design of the study; in the collection, analyses, or interpretation of data; in the writing of the manuscript, or in the decision to publish the results.

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
