# Peer review of "Extraction of Information on Trees outside Forests Based on Very High Spatial Resolution Remote Sensing Images"

_forests, doi:10.3390/f10100835_

Round 1
Reviewer 1 Report
ready to publish
Reviewer 2 Report
This is very diligent work. I particularly appreciate the inclusion of field surveys. Moreover the work is easily understandable and also of particular importance as well with respect to urgent environmental issues and practical application, as methodological in the field of automatic analysis of remote-sensing data.
This manuscript is a resubmission of an earlier submission. The following is a list of the peer review reports and author responses from that submission.
Round 1
Reviewer 1 Report
Review of Sun et al ms forests-542135.
This was a good study and well-written report; I enjoyed reading it and the findings are encouraging for future application and further development of remote methods for studying tree spatial distributions. The authors are commended for excellent English text! The title is rather long – perhaps the words after the colon could be eliminated.
My comments in general: Two major concerns. First, I think many readers will have difficulty calling these Ulmus habitats “forest”. The authors clearly point out in several places that this is sparsely wooded and may have a savanna appearance. Many readers implicitly think of “forest” as closed-canopy. I think “woodland” would be a better word; this would need to be a change in multiple locations throughout the manuscript. Second, the authors have a very nice paper here, but have not really explained why knowing tree distribution is important – I agree that many applications could benefit from knowing the spatial distribution of individual trees, but none of these are presented in the manuscript. For example, in lines 44 – 54, top of page 2, the authors tell us that is of “scientific and ecological importance” to be able to identify individual U. pumila trees – but the justification is that it could help in “developing countermeasures designed to maintain and improve the regional ecosystem stability”. How? Similar criticism on page 3, lines 133-139.
My detailed comments:
Abstract, lines 22 and 26: change “forest” to “trees”.
Abstract, lines 14 & 33: change “forest” to “woodland”.
Figure 1: the text in the figures is far too small and difficult to read.
Section 2.2.1, line 147: change to “two cameras, one that provides 1-m spatial resolution panchromatic (0.14-0.9 um) and one that provides 4-m resolution”.
Section 2.2.2, line 175: change to “we set 16 uniform test quadrats, arranged in a 4 x 4 grid, each…..”
Section 2.3, lines 182-184: First sentence could be eliminated.
Figure 3: the text in the figure is too small; difficult to read.
Section 2.3.3, lines 232-233: The second sentence in this paragraph is redundant and repeats what was said above.
Section 3.1.1, lines 270-274: I am confused – it seems to me that the Patch-max size should *NOT* be increased – it seems like increasing Patch-max will cause clumps of 2-3 trees to be treated as single trees. Please clarify.
Section 3.1.2, lines 306-317, and Figure 6: I am not sure this paragraph, and Figure 6, are necessary. I am convinced by earlier text that the ideal Sigma value is roughly 90….the figure does not make me more convinced.
Section 4 (Discussion), line 369: Unclear what is proposed….the text beginning with “It is proposed….” Is not a complete sentence.
Section 4 (Discussion), lines 373-379: Perhaps this text is obvious to most readers – surely it’s a given that 10 m resolution (or coarser) is not sufficient to delineate tree crowns that are often 2-5 m in diameter. Perhaps these lines could be eliminated.
Section 4 (Discussion), lines 413 and 416: Not clear why researchers would combine high-resolution LIDAR with this type of imagery analysis – it seems to me that the LIDAR alone would be quite sufficient for the purposes discussed in this paper.
Overall, this is still a good manuscript. I suggest minor revision, but be sure to clarify the few points in my general comments, and my confusion in section 3.1.1.
Reviewer 2 Report
The authors describe a methodology, based on remotely sensed data, for extracting the location of Ulmus Pumila. To achieve such results, the authors show a pipeline based on a multiscale automated approach that makes use of fused image feature transform.
Even if I recognize the efforts and the rigor used to perform the research. Here following the detailed list of comments.
The introduction is appropriate, since it describes the state of art; however, I might expand the main contributions of the article, since it is just mentioned at the end of the section.
The preprocessing is done to create a multiscale feature space, but much more details should be provided because:
1 - it is not automated
2 - it makes use of very well known methods that can be even removed (e.g. Formula 1,2,3).
But what makes me more skeptic is the classification method proposed.
1 - it is based on a very basic statistic that does not contribute w.r.t. the state of art
2 - the authors talk about an algorithm (line 320) without any details
3 - the comparison with the other methods is shallow. Neural net, what does it means? How is it trained ? which network has been used ?
The metric used for the comparison are not comparable. In fact, whilst the method proposed by the authors is validated by iterating the Sigma value, S.o.A. methods like Maximum Likelihood use confusion matrix that is not enough to evaluate the Neural Network performances.
Finally, I might expect from the authors a ground truth to validate their approach.